# The Genetic Elements of the Obesity Paradox in Atherosclerosis Identified in an Intercross Between Hyperlipidemic Mouse Strains

**DOI:** 10.3390/ijms26094241

**Published:** 2025-04-29

**Authors:** Mei-Hua Chen, Bilhan Chagari, Ashley M. Abramson, Lisa J. Shi, Jiang He, Weibin Shi

**Affiliations:** 1Department of Radiology and Medical Imaging, University of Virginia, Charlottesville, VA 22903, USA; sjchen@email.com (M.-H.C.); bc8he@virginia.edu (B.C.); ama4sy@virginia.edu (A.M.A.); lisajshi@gmail.com (L.J.S.); jh6qv@uvahealth.org (J.H.); 2Biochemistry & Molecular Genetics, University of Virginia, Charlottesville, VA 22908, USA

**Keywords:** obesity paradox, atherosclerosis, obesity, mice, genetic linkage

## Abstract

Overweight and obese individuals show lower mortality rates or better prognoses than those of normal weight in a variety of diseases, a phenomenon called the “obesity paradox”. An inverse association of adiposity with atherosclerosis has been observed in both humans and mice. To dissect phenotypic and genetic connections between the traits, 154 female and 145 male F2 mice were generated from an intercross between BALB/cJ and LP/J apolipoprotein E-deficient mice and fed a Western diet for 12 weeks. Atherosclerotic lesion size in the aortic root, body weight, plasma lipids, and glucose were measured, and genotyping was performed on miniMUGA SNP arrays. Quantitative trait locus (QTL) analyses on all F2 mice with sex as a covariate revealed four significant QTLs on chromosomes (Chr) 3, 6, 13, and 15 for atherosclerosis and three significant QTLs on Chr2, 7, and 15 for body weight. Chr15 QTL for atherosclerosis overlapped with one for body weight near 36 Mb. After adjusting for variation in body weight, Chr15 atherosclerosis QTL was downgraded from significant to suggestive linkage. Body weight was inversely correlated with atherosclerotic lesion sizes and accounted for more variance than a single other risk factor for atherosclerosis among F2 mice. Analysis of public data collected from two backcross cohorts revealed strong correlations between body weight and fat mass in adult mice (r ≥ 0.93; *p* ≤ 1.6 × 10^−136^). Thus, the obesity paradox in atherosclerosis is partially attributable to shared genetic components that have an opposite effect on adiposity and atherosclerosis.

## 1. Introduction

Obesity, defined as excessive fat accumulation in the body, is a risk factor for a variety of chronic diseases, including metabolic syndrome, type 2 diabetes, atherosclerosis, and cancer [1]. Body mass index (BMI), calculated as a person’s weight in kilograms divided by the square of height in meters, is a widely used indicator for defining adiposity. In a large multicenter registry of patients without known coronary artery disease undergoing coronary tomographic angiography, BMI is independently associated with the prevalence, extent, and severity of coronary artery disease [2]. Higher BMI is also independently associated with type 2 diabetes, hypertension, and dyslipidemia, all of which are major risk factors for atherosclerosis [3]. A meta-analysis of 239 prospective studies involving over ten million participants in Asia, Europe, and America shows associations of both overweight and obesity with higher all-cause mortality in the general population [4]. Obesity is associated with adipose tissue overproduction of certain adipokines and pro-inflammatory cytokines, which predispose to a state of low-grade inflammation, and with systemic metabolic disturbances [5,6,7]. The latter include insulin resistance, dyslipidemia, and endothelial dysfunction, which are risk factors for atherosclerosis [8]. Mendelian randomization is an analysis that uses genetic variants as an instrumental variable to infer the causal effect of an exposure on an outcome. This analysis has also revealed causal relationships between higher BMI and coronary artery disease [9].

Clinical observation studies have suggested a complex relationship between obesity and cardiovascular disease. Initially, it was observed that overweight and obese patients with coronary heart disease undergoing percutaneous coronary intervention had a lower incidence of major in-hospital complications, including cardiac death, compared to their normal-weight counterparts, a phenomenon called the “obesity paradox” [10]. Since then, the obesity paradox has been reported in overweight and obese patients with stable coronary heart disease [11], acute coronary syndromes [12,13], or those undergoing coronary artery bypass grafting [14]. A systematic review of 40 cohort studies on 250,152 patients with coronary artery disease revealed significantly lower total mortality and cardiovascular mortality rates in overweight individuals compared with those with a normal BMI during a mean follow-up period of 3.8 years [11]. These mortality risks were not increased in obese patients, although severely obese patients showed the highest risk for cardiovascular mortality. The protective effect of obesity has also been observed in subjects with heart failure, atrial fibrillation, and non-cardiovascular diseases [15]. As the prognosis of a disease is influenced by many factors, such as age, sex, race, severity, and treatment, more objective parameters are required to establish the causal connections between adiposity and cardiovascular disease. Atherosclerotic plaques were directly or indirectly assessed in human autopsy and coronary angiographic studies, which revealed a paradoxical inverse relationship between obesity and atherosclerosis [16,17]. In mice, body fat and atherosclerotic lesions can be more accurately measured. Inverse correlations of atherosclerotic lesion sizes with fat mass or body weight have been observed in all three mouse F2 populations thus far analyzed for associations between the traits [18,19,20]. Body weight is strongly correlated with fat mass in mice [18,21,22]. However, the genetic basis underlying the paradoxical relationship between adiposity and atherosclerosis remains unknown. Therefore, in this study, we sought to dissect phenotypic and genetic connections between the two traits in a segregating F2 population derived from two hyperlipidemic mouse strains. When overlapping QTLs for two distinct traits were found in the same cross, a causal inference test was performed to estimate the causative effect of one trait on the outcome of the other [23]. When a QTL for a particular trait was mapped in two or more crosses derived from different parental strains, available sequence and variant data were used to prioritize underlying candidate genes.

## 2. Results

Sex differences in atherosclerosis, body weight, and other risk factors. Atherosclerotic lesions in the aortic root were measured for 154 female and 145 male F2 mice after being fed a Western diet for 12 weeks. As seen in other F2 cohorts derived from *Apoe*^−/−^ strains [19,24,25,26], female mice developed larger lesions in the aorta than male F2 mice (171,219 ± 96,504 vs. 123,721 ± 68,875 µm^2^/section; *p* = 1.5 × 10^−6^; Table 1). However, there was a wide range of variation in lesion sizes among either male or female F2 cohorts: Females displayed a 32-fold difference between the mouse with the largest lesion (595,364 μm^2^) and the one with the smallest lesion (18,744 μm^2^), and males displayed a 26-fold disparity between the mouse with the largest lesion (361,448 μm^2^) and smallest lesion (14,140 μm^2^) (Figure 1A). Male and female F2 mice exhibited a wide range of overlap in aortic lesion sizes.

Male mice had heavier body weight than female mice (34.5 ± 6.2 vs. 24.4 ± 4.7; *p* = 1.25 × 10^−39^; Figure 1B), as seen in other F2 crosses derived from *Apoe*^−/−^ strains [24,26], and higher HDL cholesterol, triglyceride, and glucose levels on the Western diet under either fasting or non-fasting conditions (Table 1). Total and non-HDL cholesterol levels were comparable between the two sexes on the Western diet under either fasting or non-fasting conditions. On the chow diet, male F2 mice had higher HDL, lower total and non-HDL cholesterol, and similar triglyceride and glucose levels compared to female F2 mice.

Associations of atherosclerosis with risk factors. Associations of atherosclerotic lesions with other measures were determined with the entire F2 cohort, male and female sub-cohorts (Table 2). When all F2 mice were analyzed with a generalized linear model using sex as a covariate, atherosclerotic lesions showed inverse associations with body weight (r = −0.286, *p* = 0.013; Figure 2) and plasma triglyceride on the chow diet (r = −0.132, *p* = 0.012) but a positive association with plasma glucose on the Western diet (r = 0.076, *p* = 0.009). For the female mice, atherosclerotic lesions only displayed an inverse association with non-fasting plasma triglyceride on the chow diet (r = −0.249, *p* = 0.0019). For the male mice, atherosclerotic lesions showed an inverse association with body weight (r = −0.242, *p* = 0.003) but positive associations with non-fasting triglyceride (r = 0.20, *p* = 0.16) and glucose (r = 0.165, *p* = 0.049) and fasting glucose (r = 0.203, *p* = 0.015) on the Western diet.

Body weight is a surrogate measure of body fat in mice. Based on the r^2^ value, body weight accounted for 8.2% of the variance in atherosclerotic lesion sizes of F2 mice, the largest among the risk factors tested. Using a public dataset obtained from two female backcross populations fed a high-fat diet [26], we evaluated the relationship between body weight and fat mass in mice. This relationship was not determined in that study. At all four time points, 3, 6, 10, and 15 weeks of age, body weight showed strong positive correlations with fat mass in both crosses, with r ≥ 0.77 and *p* ≤ 7.0 × 10^−62^ (Figure 3). At the mature adult stage (10 and 15 weeks of age), the correlations between the two measures were even stronger, with r ≥ 0.93 and *p* ≤ 4.7 × 10^−157^.

QTL analysis of atherosclerosis. QTL analysis on all F2 mice with sex as a covariate revealed four significant loci on chromosomes (Chr) 3, 6, 13, and 15 and two suggestive loci on Chr15 and 17 for atherosclerotic lesion sizes (Figure 4A). Details of these QTLs, including locus name, LOD score, peak marker, 95% confidence interval (CI), mode of inheritance, high allele, and allelic effect, are shown in Table 3. When male and female F2 mice were analyzed separately, Chr3, 13, and 15 QTLs for atherosclerosis were detected in both sexes, but Chr6 and 17 QTLs were only found in the females (Figure 4C,D). The Chr6 QTL at 59.7 Mb had a significant LOD score of 4.95. This QTL is named *Ath54* to represent a novel atherosclerosis locus of mice. The Chr17 QTL replicates *Ath26* previously mapped in an intercross between DBA/2 and AKR *Apoe*^−/−^ mice [24]. The Chr13 QTL at 103.6 Mb had a significant LOD score of 3.99, replicating *Ath32* mapped in a B x H intercross [19]. The Chr3 QTL replicates *Ath51*, the proximal and distal Chr15 QTLs replicate *Ath52* and *Ath53*, respectively [27].

A genome-wide QTL scan for atherosclerosis was also performed on all F2 mice using sex as an interactive covariate (Figure 4B). Compared with the QTL scan without including sex as an interactive covariate, this scan raised the LOD score of Chr6 QTL from 4.95 to 8.34 and Chr17 QTL from 2.76 to 4.53 but barely changed the LOD scores of Chr3, 13, and 15 QTLs (Table 2). The LOD score difference between the two QTL scans with and without sex being an interactive covariate may reflect the sex effect on the strength of a specific QTL. Indeed, the Chr6 and 17 QTLs were only found in the female but not male F2 mice, while the Chr3, 13, and 15 QTLs were detected in both sexes (Figure 4C,D). The BALB allele increased atherosclerosis at Chr3, 6, and 17 QTLs, while the LP allele increased atherosclerosis at Chr13 and 15 QTLs.

QTL analysis of body weight. QTL scan on all F2 mice using sex as a covariate revealed three significant loci on Chr2, 7, and 15 and two suggestive loci on Chr1 and 14 for body weight (Figure 5A, Table 3). The Chr2 QTL was detected in both male and female F2 mice, while the Chr7 and Chr15 QTLs were only detected in females and the Chr14 QTL in males (Figure 5C,D). The Chr2 QTL at 173.2 Mb had a significant LOD score of 6.05. This QTL replicates *Adip26* mapped in C57BL/6ByJ × 129P3/J F2 mice [24] and *Mob5* mapped in CAST/Ei × C57BL/6J F2 mice for body fat [21]. The Chr7 QTL replicates *Bw1n* mapped in an intercross between Nagoya-Shibata-Yasuda (NSY) and C3H mice [28] and *Bdwtq* dissected in a subcongenic strain [29]. The Chr15 QTL at 35.4 Mb had a significant LOD score of 6.27, replicating *Bsbob5* mapped in a spontaneous obesity mouse model [30], *Dob9* mapped in CAST/Ei × C57BL/6J F2 mice [31], and *Dob3* mapped in AKR/J × SWR/J F2 mice [32].

When a genome-wide QTL scan for body weight was performed by including sex as an interactive covariate, the LOD score of Chr7 QTL rose from 4.03 to 5.36 and the Chr14 QTL from 3.51 to 4.79, while the other three QTLs showed little change in their LOD scores (Figure 5B). When QTL analysis for body weight was run separately for each sex of F2 mice, three significant QTLs on Chr2, 7, and 15 and one suggestive QTL on Chr1 were detected in the females, and two suggestive QTLs on Chr2 and 14 were detected in the males (Figure 5C,D).

Causal link between atherosclerosis and body weight. The atherosclerosis QTL, *Ath52*, overlapped with the QTL for body weight on Chr15 near 36 Mb. At the locus, the LP allele increased atherosclerosis but decreased body weight, while the BALB allele had opposite effects on the two traits. The opposing effect of this locus on atherosclerosis and body weight is concordant with the inverse correlation between the two traits observed in the F2 cohort. To evaluate the likely causal effect of body weight on atherosclerosis, a QTL scan was performed on residuals derived from the regression analysis of body weight with atherosclerotic lesions in the F2 mice. After excluding the influence from variation in body weight, the Chr15 atherosclerosis QTL was downgraded from a significant to suggestive linkage (LOD score reduced from 6.82 to 4.63) (Figure 6), and the Chr17 QTL was down in its LOD score from 4.53 to 3.93, while other QTLs were minimally affected.

Prioritization of candidate genes for significant QTL. Significant QTLs (*Dob3*, *Dob9, Mob4*) for body fat have been mapped to the Chr15 region near 35 Mb in AKR/J × SWR/J, CAST/Ei × C57BL/6J F2 crosses, and a (SPRET/Ei × C57BL/6J) × C57BL/6J N2 cross [31,32,33]. At the locus, the BALB, AKR/J, and B6 alleles increased body fat, whereas the LP, SWR/J, SPRET/Ei, and CAST/Ei alleles reduced it. Thirty-nine genes underneath the linkage peak of Chr15 QTL for body weight (5 Mb on either side of the peak marker) are polymorphic, containing multiple SNPs that are shared by the high allele strains but different from SNPs of the low allele CAST/Ei strain (Appendix A). Of them, *Ctnnd2*, *Dap*, *Marchf6*, *Cmbl*, *Sdc2*, *Cpq*, *Stk3*, *Vps13b*, *Cox6c*, *Grhl2*, *Fzd6*, *Cthrc1*, *Lrp12*, and *Zfpm2* showed associations or had functions related to atherosclerosis or obesity.

Under the linkage peak of Chr6 QTL for atherosclerosis, *Ath54*, 41 protein-coding genes were polymorphic between BALB and LP strains, including *Jazf1*, *Creb5*, *Tril*, *Cpvl*, *Chn2*, *Wipf3*, *Scrn1*, *Fkbp14*, *Plekha8*, *Mturn*, *Znrf2*, *Nod1*, *Ggct*, *Gars1*, *Crhr2*, *Inmt*, *Mindy4*, *Aqp1*, *Adcyap1r1*, *Neurod6*, *Itprid1*, *Ppp1r17*, *Pde1c*, *Lsm5*, *Avl9*, *Kbtbd2*, *Fkbp9*, *Nt5c3*, *Ppm1k*, *Herc6*, *Pyurf*, *Lancl2*, *Vopp1*, *Abcg2*, *Herc3*, *Fam13a*, *Gprin3*, *Snca*, *Mmrn1*, *Ccser1*, and *Grid2*. Of these, *Jazf1*, *Chn2*, *Nod1*, *Crhr2*, *Aqp1*, *Pde1c*, *Ppm1k*, *Abcg2*, and *Mmrn1* are functional candidates.

## 3. Discussion

This study has examined the phenotypic and genetic relationships between body weight and atherosclerosis in a large F2 cohort derived from two hyperlipidemic *Apoe*^−/−^ mouse strains. Body weight showed an inverse correlation with atherosclerotic lesion sizes, indicating the existence of the obesity paradox in atherosclerosis. Four significant QTLs on chromosomes 3, 6, 13, and 15 for atherosclerotic lesions and three significant QTLs on chromosomes 2, 7, and 15 for body weight were identified. The QTL for atherosclerosis was colocalized with the QTL for body weight on chromosome 15 near 36 Mb. After excluding the influence of variation in body weight, the atherosclerosis QTL on chromosome 15 was downgraded from significant to suggestive linkage, demonstrating the causal effect of body weight on atherosclerosis. In addition, using data from two backcross populations fed a high-fat diet, we demonstrated that body weight is a simple but reliable surrogate indicator of body fat in mice.

A major finding of this study is the negative correlation between body weight and atherosclerotic lesion sizes in the F2 mice, largely contributed by the males. An inverse association of atherosclerotic lesions with body weight or fat mass has been observed in other mouse crosses or cohorts [19,33,34]. Although the fat mass of the F2 mice was not measured in this study, the linear correlations between body weight and fat mass observed at all tested age points with two large backcrosses fed a high-fat diet indicate that body weight is near-equivalent to body fat in mice. The correlation coefficients between the two measures were extremely high for adult mice, with the r values exceeding 0.9.

In this study, we observed the colocalization of QTLs for atherosclerosis (*Ath52*) and body weight (*Bsbob5*) on chromosome 15. This colocalization provided a statistical means to decipher causal connections between the two traits. By excluding the effect of variation in body weight, we demonstrated the dependence of the atherosclerosis QTL on body weight. Indeed, after adjustment for body weight, the atherosclerosis QTL *Ath52* was downgraded from significant to suggestive linkage, indicating a causal effect of body weight on atherosclerosis.

At the overlapping QTLs on chromosome 15, the BALB allele increased body weight but reduced atherosclerosis, while the LP allele had opposite effects. The opposite allelic effect from the same QTL aligns with the inverse correlation between the two traits observed with the F2 mice. Thus, a pleiotropic gene or two different but closely linked genes with an opposite effect on atherosclerosis and obesity underlie the observed obesity paradox in atherosclerosis.

The chromosome 15 QTL for body weight identified in this cross was coincident with *Dob3* for body fat mapped in AKR/J × SWR/J F2 mice [32], *Dob9* mapped in CAST/Ei × C57BL/6J F2 mice [32], and *Mob4* for body fat mapped in (SPRET/Ei × C57BL/6J) × C57BL/6J N2 mice [33]. When QTLs for the same trait are mapped to the same chromosomal region in multiple crosses derived from different parental strains, genomic sequence variant data for these strains allows us to prioritize positional candidate genes. Thirty-nine candidate genes at the QTL were found to contain one or more variants that were shared by the high allele strains (BALB, B6, AKR) but different from the ones of the low allele strains (LP, SWR, Cast/Ei). This analysis is based on the finding that 97% of the genetic variants between common inbred mouse strains are ancestral, and thus, the causal gene of a QTL most likely contains genetic variant(s) shared among the founder strains [20]. CAST/Ei is a wild-derived mouse strain that is composed mainly of *M. m. castaneus* genomic segments, while the classic inbred strains BALB, B6, AKR, and LP are derived mainly from *M. m. domesticus*. Thus, for most candidate genes, the haplotype-based analysis was performed without including Cast/Ei mice.

This study revealed sex differences in atherosclerosis and related risk factors, including body weight, plasma lipids, and glucose, between male and female F2 mice in the same cross derived from the two *Apoe*^−/−^ strains. As seen in other crosses [19,24,26,35,36], female F2 mice developed larger atherosclerotic lesions than their male counterpart. Male F2 mice had larger body weight, higher plasma triglycerides, HDL cholesterol, and glucose levels than their female counterpart on the Western diet. The finding that male F2 mice had larger body weight but developed smaller atherosclerotic lesions relative to their female counterpart also supports the existence of the obesity paradox in atherosclerosis. Body weight, together with plasma triglycerides and glucose on the Western diet, contributed to the variance in atherosclerotic lesion sizes among male F2 mice, whereas only plasma triglycerides on the chow diet contributed to the variance in lesion sizes among female F2 mice. The coefficient of determination, *r*^2^, in linear regression estimates the proportion of the variance in one variable that is accounted for by the other. Based on the *r*^2^ values, body weight accounted for more variance than a single other risk factor in atherosclerotic lesion sizes, at 8.2% among all F2 mice and at 5.9% among male F2 mice. Plasma triglycerides and glucose explained 2.7% and 4.1% of the variance in lesion sizes of male F2 mice, respectively. Among the female F2 mice, plasma triglycerides on the chow diet explained 6.2% of the variance in lesion sizes, displaying a protective effect against atherosclerosis. Negative associations between plasma triglycerides and atherosclerosis have also been observed in other crosses [19,37,38].

Although higher HDL could protect the male mice against atherosclerosis, it showed no association with atherosclerotic lesion sizes in either male or female F2 mice. Non-HDL cholesterol showed neither sex difference nor association with atherosclerotic lesions in the F2 mice. Poor associations of non-HDL with atherosclerotic lesion sizes have been found in other mouse crosses [19,38]. An explanation for the sex discrepancy in the correlation of atherosclerotic lesions with blood glucose and triglycerides is that male *Apoe*^−/−^ mice develop more significant glucose intolerance and type 2 diabetes on the Western diet [39], as reflected by higher glucose and triglycerides in male than female F2 mice.

We identified a significant QTL, *Ath54*, for atherosclerosis on chromosome 6 at 59.7 Mb in the female F2 mice. The homozygous BALB allele was associated with increased atherosclerotic lesion sizes. This QTL is distinct from *Artles* at 126 Mb, mapped in a B6 × Cast/Ei F2 cross [40], and *Athsq2* at 134 Mb, mapped in a (MOLF/Ei x B6) x B6 *Ldlr*^−/−^ N2 cross [41]. Potential candidate genes include *Creb5*, *Cpvl*, *Chn2*, *Nod1*, *Pde1c*, *Ppm1k*, *Vopp1*, *Abcg2*, *Herc*, *Fam13a*, *Ccser1*, and *Grid2*. These genes are highly polymorphic between the parental strains or are associated with coronary artery disease and its risk factors.

Conclusions. We have observed the paradoxical association of body weight, an indicator of fatness, with atherosclerotic lesions in a segregating F2 mouse population. The discovery of coincident QTLs with an opposite allelic effect on atherosclerosis and body weight provides an explanation for the paradoxical relationship between obesity and atherosclerosis. It is intriguing to determine whether a pleiotropic gene or two different but linked genes underlie the overlapping QTLs for body weight and atherosclerosis. As the known adipose tissue-derived anti-atherogenic genes do not reside within the confidence interval of *Ath54*, fine mapping of this chromosome 15 region may uncover new atheroprotective molecules and pathways.

Limitations of the present study. There are a few limitations with this study: First, only atherosclerotic lesions in the aortic roots were measured. There is regional variation in genetic control of atherosclerosis in hyperlipidemic mice; thus, the genetic factors driving aortic atherosclerosis may be distinct from those associated with atherosclerosis in other locations, like the carotid arteries [37]. Second, the fat mass of mice was not measured, although we showed its strong correlation with body weight from analysis of a previously published dataset. Third, the food intake of mice was not measured. The overconsumption of high-fat/high-calorie foods is a main contributor to obesity in humans [42].

## 4. Methods

**Mice**. F2 mice were generated from an intercross between male LP/J (LP) and female BALB/cJ (BALB) *Apoe^−/−^* mice as reported [20]. Mice were weaned onto a rodent chow diet and, at 6 weeks of age, transferred to a Western diet (TD 88137, Envigo, Madison, USA) containing 42 kca% fat, 42.7 kcal% carbohydrates, 15.2 kcal% protein, and 1.5% cholesterol. Non-fasting blood was collected once immediately before the start of the Western diet and once after 11 weeks on the diet. Fasting blood was collected, and body weight was measured after an overnight fast at the end of 12 weeks of Western diet feeding. Blood was collected from the retro-orbital veins with mice being anesthetized via isoflurane inhalation, and plasma was prepared as reported [43]. All procedures were conducted according to the protocols approved by the Institutional Animal Care and Use Committee (protocol #: 3109).

**Phenotypic analyses**. Atherosclerotic lesion sizes in the aortic root of F2 mice were measured as reported [27]. Briefly, the vasculature was perfusion-fixed with 10% formalin via the left ventricle. The aortic root and adjacent cardiac tissue were harvested, embedded in Tissue-Tek OCT compound, and cross-sectioned at 10 μm thickness. Every 5th section was collected and stained with oil red O and hematoxylin. Aortic lesion areas were measured using Zeiss AxioVision 4.8 software. Five sections with the largest lesion areas were averaged for each mouse, and this average was used for statistical analysis. Plasma cholesterol and triglyceride concentrations were measured using enzymatic assays as reported [43]. HDL cholesterol was measured after precipitation of other lipoproteins with a phosphotungstate-magnesium reagent (Wako Diagnostics). Non-HDL cholesterol was calculated by subtracting HDL cholesterol from total cholesterol concentrations. Plasma glucose was measured using a Sigma assay kit (Cat. # GAHK20) as reported [43].

**Genotypic analysis**. DNA was extracted from the tail of mice and used for genotyping at Neogen (Lansing, MI, USA) with miniMUGA arrays containing 11,000 SNP probes. Parental and F1 DNA served as controls on each array. SNP markers that showed unexpected genotypes for control samples or deviation from the Hardy–Weinberg proportion were excluded. Genotyping errors were also checked using the “calc errorlod” function of R/qtl. After filtration, 2594 informative SNPs remained and were used for QTL analysis.

**Statistical analysis**. QTL analysis was performed using R/qtl as reported [28]. Genome-wide thresholds for significant (*p* < 0.05) and suggestive (*p* < 0.63) linkage with a trait were determined by permutation tests [44,45]. One thousand permutations were run in a 1 Mb interval across the entire genome and with the EM (Expectation–Maximization) algorithm for each trait. As atherosclerotic lesions and body weight were significantly different between male and female mice, QTL scans were run separately for each sex. To increase statistical power, all F2s of both sexes were included in QTL scans using sex as a covariate or an interactive covariate.

The Student’s *t*-test was used to determine differences between male and female F2 mice in quantitative traits. Linear regression analysis was performed to determine associations between continuous measures. For measures that were significantly different between male and female mice, linear regression analysis was performed with adjustment for sex; *p*-values of ≤0.05 were considered statistically significant.

**Causal inference based on QTL colocalization**. When QTLs for two distinct traits were mapped to the same chromosomal region, potential causal connections between the traits were evaluated using a statistical method as described [23]. Briefly, residuals were generated from linear regression analysis of two associated traits and were then subjected to a genome-wide QTL scan with the same algorithm as used for mapping the coincidental QTLs. The QTLs yielded from the residual variation in one trait should be independent of the variation in the other trait.

**Candidate gene prioritization**. Bioinformatics resources were used to prioritize candidate genes for significant QTLs of interest. Genome sequence variants were retrieved through the Mouse Genome Informatics and Ensembl Genome Browser. Potential candidate genes in the confidence interval of a significant QTL were those containing one or more variants that were shared by high allele strains but were different from the ones of low allele strains, as described [37]. Of them, those containing one or more non-synonymous SNPs or SNP(s) in upstream regulatory regions were considered probable candidate genes as described [37].

**Body weight versus body fat**. Association between body weight and body fat was evaluated using publicly available data obtained from two N2 mouse cohorts, NZO/HI × F1(NZO/HI × C3HeB/FeJ) and NZO/HI × F1(NZO/HI × 129P2/OlaHsd) [26]. The former cohort consisted of 310 female N2 mice, and the latter contained 307 female N2 mice. These N2 mice were weaned at 3 weeks of age onto a high-fat diet containing 45 kcal% fat, 20 kcal% protein, and 35 kcal% carbohydrates (D12451 Research Diets Inc., New Jersey, USA) and kept on the diet thereafter [26]. Body weight and body fat of mice were measured at 3, 6, 10, and 15 weeks of age. Fat mass was measured by nuclear magnetic resonance spectroscopy (EchoMRI™-100 System, Echo Medical Systems, Houston, TX, USA).

## Figures and Tables

**Figure 1 ijms-26-04241-f001:**
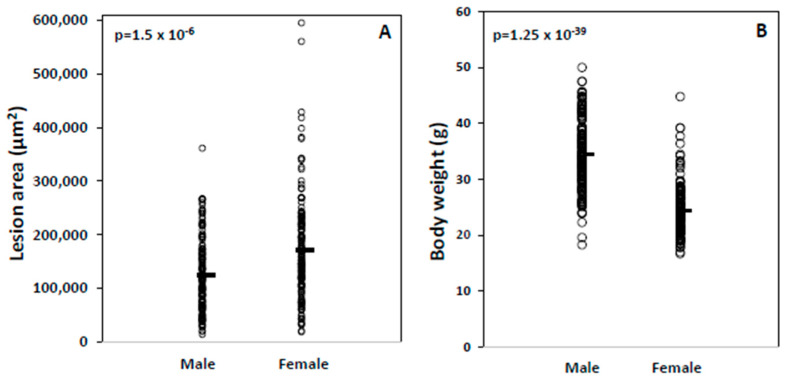
Atherosclerotic lesions (**A**) and body weight (**B**) of male and female F2 mice after being fed 12 weeks of a Western diet. Each symbol represents an individual mouse. The short horizontal lines denote the mean values of each group. The *p*-values derived from the Student’s *t*-test are shown.

**Figure 2 ijms-26-04241-f002:**
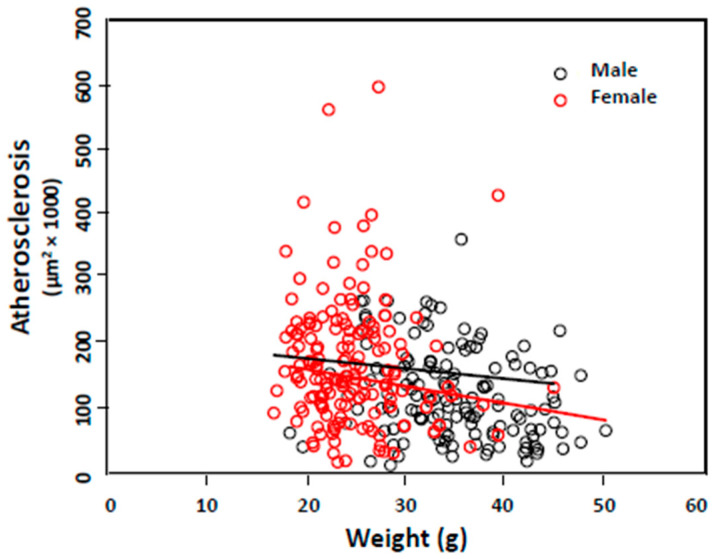
Associations of body weight with atherosclerotic lesion sizes in male and female F2 mice. The regression line was determined using the linear regression model for each sex separately. Each circle represents the values of an individual F2 mouse.

**Figure 3 ijms-26-04241-f003:**
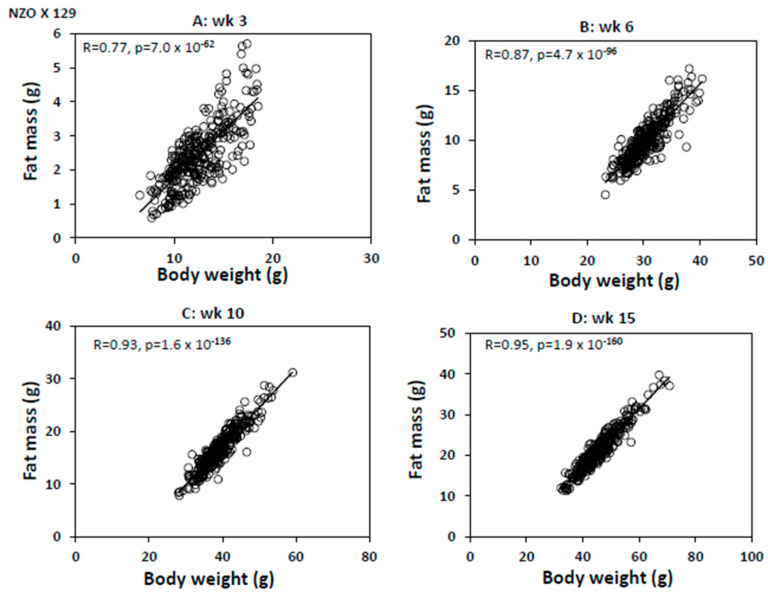
Correlations between body weight and fat mass in two female backcross mouse populations, NZO × 129 mice (**A**–**D**) and NZO × C3H (**E**–**H**), at 3, 6, 10, and 15 weeks of age. Mice were fed a high-fat diet after being weaned at 3 weeks of age. The regression line and *p* and r values were determined by linear regression analysis. Each dot represents an individual mouse.

**Figure 4 ijms-26-04241-f004:**
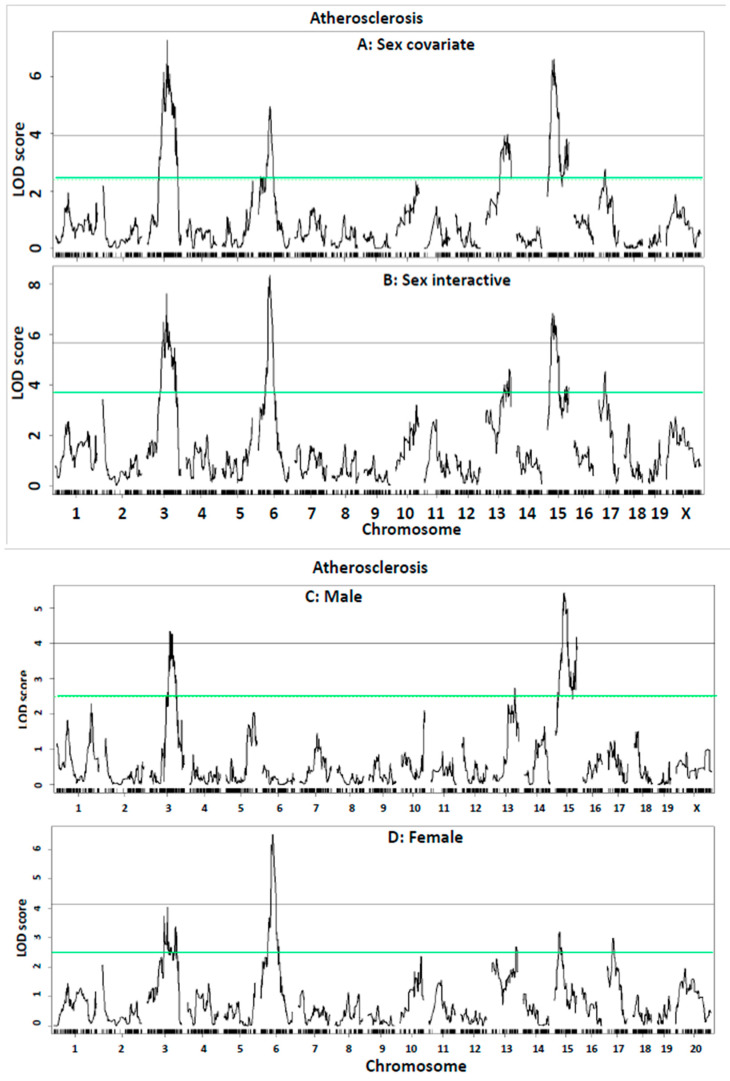
Genome-wide QTL scans for atherosclerotic lesions. (**A**) All F2 mice were included in QTL analysis using sex as a covariate; (**B**) all F2 mice were included in QTL analysis using sex as an interactive covariate; (**C**) only male F2 mice were included in QTL analysis; and (**D**) only female F2 mice were included in QTL analysis. The X-axis denotes chromosomes 1 through X, and the Y-axis represents the LOD score. The horizontal green and black lines indicate the genome-wide thresholds for suggestive (*p* < 0.63) and significant linkage (*p* < 0.05), respectively. Each vertical short black line sitting on the X-axis represents a genetic marker.

**Figure 5 ijms-26-04241-f005:**
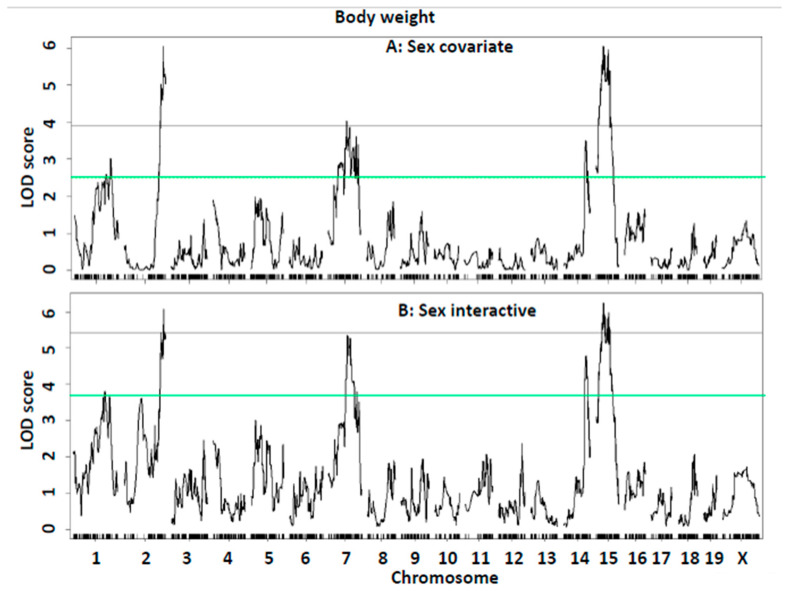
Genome-wide QTL scans for body weight. (**A**) All F2 mice were included using sex as a covariate; (**B**) all F2 mice were included using sex as an interactive covariate; (**C**) only male F2 mice were included; and (**D**) only female F2 mice were included. The X-axis indicates chromosomes 1 through X, and the Y-axis represents the LOD score.

**Figure 6 ijms-26-04241-f006:**
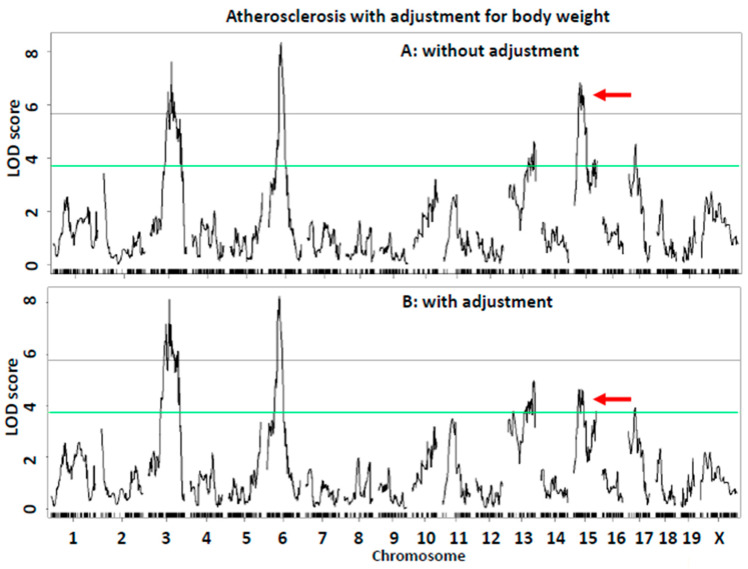
QTL analysis to evaluate the dependence of atherosclerosis QTLs on body weight. (**A**) Genome-wide scan for atherosclerosis without adjusting for body weight. (**B**) Genome-wide scan for atherosclerosis after adjustment for body weight. Note the downgradation of chromosome 15 QTL for atherosclerosis from significant to suggestive linkage (pointed by red arrow).

**Table 1 ijms-26-04241-t001:** Phenotypic variation in atherosclerosis and related risk traits between male and female F2 mice.

Phenotype	Male	SD	Female	SD	*p* Value
Weight	34.5	6.2	24.4	4.7	1.25 × 10^−39^
Coat color	1.3	0.9	1.3	0.9	0.69
Atherosclerosis	123,721	68,875	171,219	96,504	1.53 × 10^−6^
Total Cholesterol (nonfast chow)	355.7	87.0	415.8	105.0	1.42 × 10^−7^
Non-HDL (nonfast chow)	317.0	86.7	390.4	103.9	1.57 × 10^−10^
Triglycerides (nonfast chow)	93.8	33.0	95.8	33.0	0.61
HDL (nonfast chow)	38.7	14.0	25.4	11.9	1.3 × 10^−16^
Glucose (nonfast chow)	277.4	44.4	281.3	42.2	0.44
Total Cholesterol (nonfast Western)	1206.2	276.3	1184.9	225.6	0.47
Non-HDL (nonfast Western)	1176.5	277.3	1168.3	228.7	0.78
Triglycerides (nonfast Western)	162.1	72.8	120.4	45.8	1.29 × 10^−8^
HDL (nonfast Western)	29.6	12.4	16.6	9.8	2.36 × 10^−20^
Glucose (nonfast Western)	391.4	134.4	313.4	92.8	1.91 × 10^−8^
Total cholesterol (fast Western)	1232.5	480.7	1192.9	373.2	0.43
Non-HDL (fast Western)	1204.8	483.1	1177.5	373.4	0.59
Triglycerides (fast Western)	185.6	107.4	132.9	62.9	6.86 × 10^−7^
HDL (fast Western)	27.7	20.2	15.5	10.8	9.3 × 10^−10^
Glucose (fast Western)	368.3	168.3	296.0	121.1	3.42 × 10^−5^

The F2 cohort consisted of 154 female and 144 male mice. Results are means ± SD. The *p*-values were derived from the Student’s *t*-test. Unit: weight in grams; coat color with white being 0, light brown 1, dark brown 2, agouti 3, and dark brown 4; plasma lipids and glucose in mg/dL; atherosclerotic lesion area, µm^2^/section. Traits with a statistical significance between male and female mice are denoted by an underline.

**Table 2 ijms-26-04241-t002:** Association of atherosclerotic lesion sizes with risk factors in F2 mice.

Trait	R (♂ + ♀)	P (♂ + ♀)	R (♀)	P (♀)	R (♂)	P (♂)
Body weight	**−0.286**	**0.013**	−0.076	0.351	**−0.242**	**0.0036**
Coat color	−0.056	0.366	−0.064	0.429	0.038	0.659
LDL (chow, nonfast)	0.083	0.777	0.038	0.644	0.019	0.821
HDL (chow, nonfast)	−0.069	0.249	−0.088	0.277	0.045	0.597
Triglycerides (chow, nonfast)	**−0.132**	**0.012**	**−0.249**	**0.0019**	0.004	0.962
Glucose (chow, nonfast)	0.013	0.980	0.087	0.281	0.130	0.121
LDL (Western, nonfast)	0.024	0.634	0.041	0.611	0.117	0.165
HDL (Western, nonfast)	−0.130	0.867	−0.049	0.545	−0.078	0.353
Triglycerides (Western, nonfast)	−0.031	0.279	0.071	0.381	**0.200**	**0.016**
Glucose (Western, nonfast)	−0.011	0.154	0.014	0.860	**0.165**	**0.049**
LDL (Western, fast)	0.048	0.315	0.065	0.423	0.056	0.510
HDL (Western, fast)	−0.139	0.432	−0.133	0.102	0.014	0.871
Triglycerides (Western, fast)	−0.005	0.171	0.057	0.483	0.118	0.164
Glucose (Western, fast)	**0.076**	**0.009**	0.120	0.137	**0.203**	**0.015**

Associations of atherosclerotic lesions with various risk factors were determined with a generalized linear model adjusted by sex when both sexes of F2 mice were analyzed. ♂ indicates male, ♀ female; significant correlations are highlighted in bold. Unit: body weight: grams; coat color graded into white (0), light brown (1), dark brown (2), and agouti (3); plasma lipid and glucose levels: mg/dL.

**Table 3 ijms-26-04241-t003:** Suggestive and significant QTLs for atherosclerosis and body weight mapped in F2 mice derived from LP and BALB *Apoe*^−/−^ mice.

Locus Name	Chr	Peak (Mb)	95% CI (Mb)	SNP ID	LOD (♂ + ♀)	Allele Effect			Allele Effect				
						BB (♀)	H (♀)	LL (♀)	BB (♂)	H (♂)	LL (♂)	Mode of Inheritance	High Allele
Sex covariate													
** *Ath51* **	**3**	**93.8**	**74.8–116.8**	**UNC5770722**	**7.28**	212,840 ± 116,611	171,698 ± 92,904	128,666 ± 56,805	158,888 ± 69,998	116,758 ± 71,078	100,806 ± 45,430	Additive	BB
** *Ath54* **	**6**	**59.7**	**51.7–67.7**	**gUNC11198411**	**4.95**	235,257 ± 122,379	146,388 ± 68,344	148,862 ± 79,622	130,516 ± 72,764	123,900 ± 69,539	112,808 ± 59,829	Recessive/Additive	BB
** *Ath32* **	**13**	**103.6**	**71.6–116.6**	**S1L134123348**	**3.99**	154,509 ± 80,705	162,087 ± 85,186	207,473 ± 123,053	98,310 ± 54,368	121,464 ± 72,885	153,892 ± 62,109	Additive	LL
** *Ath52* **	**15**	**36.7**	**23.4–44.8**	**gUNCHS040036**	**6.60**	124,002 ± 58,067	186,619 ± 106,087	185,455 ± 95,362	70,087 ± 49,812	129,263 ± 72,024	146,095 ± 55,165	Dominant/Additive	LL
*Ath53*	15	92	80–100	UNC26123163	3.9	136,612 ± 80,040	179,935 ± 96,562	182,968 ± 104,654	91,311 ± 59,418	127,199 ± 67387	147,370 ± 69,551	Dominant/Additive	LL
*Ath26*	17	32.3	6.3–65.3	gUNC27802572	2.76	180,256 ± 86,546	188,275 ± 103,668	124,302 ± 72,397	151,835 ± 74,658	118,809 ± 65,059	115,870 ± 68,972	Dominant	BB
Sex interactive-covariate													
** *Ath51* **	**3**	**93.8**	**73.8–130.8**	**UNC5770722**	**7.62**	212,840 ± 116,611	171,698 ± 92,904	128,666 ± 56,805	158,888 ± 69,998	116,758 ± 71,078	100,806 ± 45,430	Additive/Recessive	BB
** *Ath54* **	**6**	**59.7**	**52.7–65.7**	**gUNC11198411**	**8.34**	235,257 ± 122,379	146,388 ± 68,344	148,862 ± 79,622	130,516 ± 72,764	123,900 ± 69,539	112,808 ± 59,829	Recessive/Additive	BB
*Ath32*	13	111.6	71.6–119.8	SBC134492977	4.6	163,038 ± 85,089	153,337 ± 79,825	224,734 ± 126,127	99,580 ± 57,002	124,853 ± 72,461	144,358 ± 64,826	Recessive/Additive	LL
** *Ath52* **	**15**	**27.8**	**22.7–45.7**	**gUNCHS040036**	**6.82**	121,464 ± 57,122	191,388 ± 107,449	184,031 ± 91,425	80,325 ± 56,562	130,859 ± 72,157	143,704 ± 56,838	Dominant/Additive	LL
*Ath53*	15	92	80–100	UNC26123163	3.9	136,612 ± 80,040	179,935 ± 96,562	182,968 ± 104,654	91,311 ± 59,418	127,199 ± 67387	147,370 ± 69,551	Dominant/Additive	LL
*Ath26*	17	32.3	3.3–51.3	gUNC27802572	4.53	180,256 ± 86,546	188,275 ± 103,668	124,302 ± 72,397	151,835 ± 74,658	118,809 ± 65,059	115,870 ± 68,972	Dominant	BB
Body weight: Sex covariate													
**Bsbob5, Dob3, Dob9**	**15**	**35.4**	**23.4–62.7**	**gUNC150117369**	**6.07**	27.2 ± 5.2	25.4 ± 4.0	22.6 ± 4.2	36.7 ± 6.6	34.8 ± 5.9	32.2 ± 5.6	Additive	BB
*Bodwt1, Fatq1*	1	161.4	3.4–172.4	S3C016445465	3.02	23.3 ± 3.1	24.3 ± 4.2	25.4 ± 5.9	31.5 ± 5.7	35.2 ± 6.3	35.7 ± 5.9	Additive/Dominant	LL
**Adip26, Mob5**	**2**	**173.4**	**163.2–181.8**	**gJAX00103339**	**6.05**	26.9 ± 5.9	24.2 ± 3.9	22.6 ± 3.5	37.0 ± 6.5	34.2 ± 6.2	32.8 ± 5.6	Additive	BB
**Bw1n, Bdwtq**	**7**	**83.1**	**46.1–135.1**	**SX1073328646**	**4.03**	23.9 ± 3.9	23.2 ± 3.5	27.6 ± 6.0	32.6 ± 5.5	34.8 ± 6.1	35.9 ± 7.0	Recessive/Additive	LL
**Bodwtq11**	14	103.1	99.7–115.7	gUNCHS039200	3.51	23.3 ± 4.2	25.5 ± 5.0	23.5 ± 4.1	34.9 ± 6.0	35.6 ± 5.9	31.0 ± 6.3	Heterosis/Dominant	H/BB
Sex interactive-covariate													
**Bsbob5, Dob3, Dob9**	**15**	**35.4**	**23.7–62.7**	**gUNC150117369**	**6.27**	27.2 ± 5.2	25.4 ± 4.0	22.6 ± 4.2	36.7 ± 6.6	34.8 ± 5.9	32.2 ± 5.6	Additive	BB
*Bodwt1, Fatq1*	1	141.4	83.4–170.1	mbHkupUNC010301903	3.81	23.2 ± 3.0	24.1 ± 4.3	25.9 ± 5.9	31.9 ± 5.5	35.4 ± 6.5	34.7 ± 5.7	Additive	LL
**Adip26, Mob5**	**2**	**173.4**	**159.6–181.8**	**c2.loc169**	**6.09**	23.3 ± 3.1	24.3 ± 4.2	25.4 ± 5.9	31.5 ± 5.7	35.2 ± 6.3	35.7 ± 5.9	Additive/Dominant	LL
**Bw1n, Bdwtq**	**7**	**83.2**	**77.4–110.1**	**SX1073328646**	**5.36**	23.9 ± 3.9	23.2 ± 3.5	27.6 ± 6.0	32.6 ± 5.5	34.8 ± 6.1	35.9 ± 7.0	Recessive/Additive	LL
Bodwtq11	14	105.6	100.2–110.7	gUNC24672839	4.79	23.3 ± 4.2	25.5 ± 5.0	23.5 ± 4.1	34.9 ± 6.0	35.6 ± 5.9	31.0 ± 6.3	Heterosis/Dominant	H/BB

Significant QTLs and LOD scores are highlighted in bold. Chr, chromosome; CI, confidence interval; ♂, male; ♀, female. BB: homozygous BALB allele; LL: homozygous LP allele; H: Heterozygous for both BALB and LP alleles. Unit for atherosclerotic lesion: µm^2^/section; plasma lipid and glucose concentration: mg/dL; body weight: g. Values for allelic effect are expressed as means ± SD.

## Data Availability

All data reported in this article are Appendix A.

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
