# Peer review of "The Genetic Elements of the Obesity Paradox in Atherosclerosis Identified in an Intercross Between Hyperlipidemic Mouse Strains"

_ijms, 2025, doi:10.3390/ijms26094241_

Round 1
Reviewer 1 Report
Comments and Suggestions for Authors
This manuscript provides solid evidence of the association between body weight, atherosclerosis, and sex. The data is consistent with published data showing higher plaque size in females than in male mice. Genes of interest were also identified.
I have only a few comments.
- For Table 1, the units of the measurements should be added. For example, atherosclerosis should be m Also, instead of atherosclerosis, it should be plaque size.
- Page 7, lines 191-195. Is this paragraph part of a figure legend?
- It is puzzling that fat mass was not measured for the experimental animals. Instead, correlations between fat mass and body weight were analyzed from other published data.
- In the discussion section, pages 393-394, the claim on body fat should be changed by body weight since fat mass was not measured in this study.
- A limitation section should be added. For example, only the aortic roots were measured. Plaque could also differ in other sections of the aorta. Fat mass was not measured. Food intake was not measured, etc…
- Figures should be revised.
- For example, in Figure 1A, there is a “.5” on the bottom right of panel A. In the same panel, one data point overlaps the upper line. Tick marks should be added to the Y-axis. The lines for the graphs are of different thicknesses. These comments also apply to other figures.
- Figure 2, add tick marks for the Y-axis and -axis.
- In Figure 3, there is a line at the top of the Figure, and something was cut off at the bottom of the figure. The same is true for Figures 4, 5, and 6.
Author Response
- Comment: For Table 1, the units of the measurements should be added. For example, atherosclerosis should be m Also, instead of atherosclerosis, it should be plaque size.
Response: The units for the measured are described under the table.
- Comment: Page 7, lines 191-195. Is this paragraph part of a figure legend?
Response: No, it’s not.
- Comment: It is puzzling that fat mass was not measured for the experimental animals. Instead, correlations between fat mass and body weight were analyzed from other published data.
Response: Although fat mass and body weight of two mouse crosses have been published, the relationship between the two measures have not been defined. The latter has clearly demonstrated that body weight is a surrogate of fat mass in adult mice.
- Comment: In the discussion section, pages 393-394, the claim on body fat should be changed by body weight since fat mass was not measured in this study.
Response: Revised.
- Comment: A limitation section should be added. For example, only the aortic roots were measured. Plaque could also differ in other sections of the aorta. Fat mass was not measured. Food intake was not measured, etc…
Response: We agree with the reviewer’s comment and have made the suggested revision.
- Comment: Figures should be revised.
- For example, in Figure 1A, there is a “.5” on the bottom right of panel A. In the same panel, one data point overlaps the upper line. Tick marks should be added to the Y-axis. The lines for the graphs are of different thicknesses. These comments also apply to other figures.
- Figure 2, add tick marks for the Y-axis and -axis.
- In Figure 3, there is a line at the top of the Figure, and something was cut off at the bottom of the figure. The same is true for Figures 4, 5, and 6.
Response: Revised.
Reviewer 2 Report
Comments and Suggestions for Authors
Reviewer 1
The authors conducted a study related to the correlations between genetic factors involved in the obesity paradox in atherosclerosis in mice with hyperlipidemia.
I believe that the article is original, and the results are presented in a clear manner, respecting scientific rigor.
I would like to make a suggestions to the authors:
- Suggest adding a paragraph related to the limitations of the study.
- Please mention the future prospects of the research and how you see the importance of the study correlated with a possible applicability in human research studies?
- I also suggest adding a paragraph with conclusions.
- Please modify the article according to the specific IJMS template (1. Introduction. 2. Results. 3. Material and method. 4. Discussion. 5. Conclusions)
Thank you for the opportunity to evaluate this article.

Author Response
- Comment: Suggest adding a paragraph related to the limitations of the study.
Response: Amended.
- Comment: Please mention the future prospects of the research and how you see the importance of the study correlated with a possible applicability in human research studies?
Response: Amended.
- Comment: I also suggest adding a paragraph with conclusions.
Response: Amended.
- Comment: Please modify the article according to the specific IJMS template (1. Introduction. 2. Results. 3. Material and method. 4. Discussion. 5. Conclusions)
Response: Amended.